# Health inequity on access to services in the ethnic minority regions of Northeastern Myanmar: a cross-sectional study

Kun Tang,[1] Yingxi Zhao,[2,3] Bolun Li,[2] Siqiao Zhang,[4] Sung Hoon Lee[5]

KT and YZ contributed equally.

[1]Department of Global Health, School of Public Health, Peking University, Beijing, China
[2]Institute for Medical Humanities, Peking University Health Science Center, Beijing, China
[3]Department of Global Health, University of Washington, Seattle, Washington, USA
[4]School of Public Health, Peking University, Beijing, Beijing, China
[5]School of International Studies, Peking University, Beijing, China

**Correspondence to**
Dr Kun Tang;
tangkun@hsc.pku.edu.cn

## ABSTRACT

**Objectives** To evaluate health inequity on access to services in the ethnic regions of Northeastern Myanmar from three points of analysis: geographic barrier, gender-based disparity and financial burden of health services.

**Setting** A multistage-stratified random cluster survey was conducted in Shan State Special Region 2 and Eastern Shan State Special Region 4 of Northeastern Myanmar in 2016, including a total number of 774 households.

**Participants** A total number of 4235 participants were recruited during the survey.

**Primary and secondary outcome measures** Geographic distance, gender, household income and inpatient/outpatient service utilisation.

**Results** The study results showed that residents living within 5 km of any form of healthcare facilities paid more outpatient visits (90.06 visits per thousand population) in the past 2 weeks, compared with those living 5–20 km and over 20 km (54.84 and 54.02 per thousand population, respectively) from healthcare facilities. A similar trend with little significant differences was found for inpatient service use. Regarding household income, adults with an annual household income of above US$720 were more likely to seek outpatient service (OR=1.43, 95% CI 0.98 to 2.10) compared with those with an annual income of <US$360. After adjusting for other covariates, female adults were less likely to seek inpatient treatment (OR=0.55, 95% CI 0.35 to 0.84) and outpatient services (OR=0.86, 95% CI 0.64 to 1.15) than male adults.

**Conclusions** Geographic barrier, gender-based disparity and financial burdens were identified as key causes that significantly restrict ethnic people's access to healthcare facilities. The study concludes that tackling health inequity in Northeastern Myanmar ethnic regions requires an improved primary healthcare system, proper financial protection mechanisms and a special focus on women.

## INTRODUCTION

Since Myanmar's signing of the Panglong Agreement in 1947 and its independence in 1948, the ethnic minority regions of Northeastern Myanmar have been torn apart by conflicts between the central government and ethnic armed organisations that seek full local autonomy. Continuous armed conflicts in Kachin, northern Shan and other states have led to the destruction of more than 200 villages and the internal displacement of some 120 000 civilians in the past years.[1] Despite the 21st Century Panglong Conference led by Aung San Suu Kyi in September 2016, which aimed to stop regional conflicts and achieve peace that has eluded the country for the past three-quarters of a century, violence in the ethnic regions has continued regardless. Recent air strikes and armed disputes across Shan State's Muse and Kutkai townships in November 2016 again resulted in dozens of casualties, with 2600 people displaced and over 3000 people forced to flee across the border into China.[2] The ongoing conflicts have led to a notoriously weak health status in the region: a 2014 malaria survey along the Myanmar–China border area suggested an incidence rate of 8.70 per thousand,[3] almost triple the level of Myanmar's national average,[4] which is all the more striking given Myanmar's already problematic health situation compared with other Southeast Asian countries in the Greater Mekong Subregion.

Health equity has always been a key focus of public health at all levels, while issues around health equity in the conflict-ridden states of Myanmar have been especially difficult to tackle due to constraints in data. Besides nationwide health inequity, armed

conflicts may also be a primary factor behind poor health status of the country, as evidenced by limited coverage of preventive intervention and high disease mortality in areas dominated by violence.[5] The infant and under-five mortality rate in Eastern Myanmar conflict regions were 94.20 and 141.90 per thousand live births in 2013, respectively, around double to triple the national average statistics of Myanmar (41.90 and 53.50 per thousand live birth in 2013, respectively),[6] and similar to the levels in Somalia.[7] Various socioeconomic determinants potentially contribute to health inequity in Myanmar's conflict areas, such as geographic barriers in access to healthcare facilities, gender inequality, displacement status, financial barrier, as well as limited health management and service capacity.[8]

### Strengths and limitations of this study

► The study represents the largest and one of the few population-representative studies of health inequity targeted in Myanmar–China border area.
► The cross-sectional nature of the study does not allow for causal inference to provide more robust evidence.
► Potential interviewer bias and respondent's recall bias may exist.

Despite deteriorating health status in the Myanmar–China border area, the issue of health inequity in Northeastern Myanmar has largely been unnoticed in the literature. The existing literature mostly focused on the Thai–Myanmar border area, with different social and political context, ethnicity composition, economic condition and health status. Hence, this study aims to analyse health inequity in the ethnic minority regions of Northeastern Myanmar, the conflict regions of Shan State in particular, with data derived from a population-representative cross-sectional survey conducted in 2016. After evaluating the level of health service utilisation in the study regions and analysing its association with various sociodemographic indicators, this study then examines the region's high health inequity mainly from three aspects: geographic barrier, gender-based disparity and financial burden of health services.

## METHODS
### Study design
A survey using multistage-stratified random cluster sampling scheme was conducted in Shan State Special Region 2 (Wa region) and Eastern Shan State Special Region 4 (SR4) of Northeastern Myanmar in March 2016. To ensure population representativeness, randomisation procedures were applied through multistage sampling: at the first stage of sampling, 75 village-level units were randomly chosen with probability proportional to scale from a sampling frame containing all village-level units under seven townships in Wa region and SR4. At the second stage, all households in a village were listed, from which 10 households were randomly chosen within each village using a geographically randomised approach—after one household was picked, every other household was chosen in a geographically clockwise sequence. A total number of 774 households and 4235 participants were eventually recruited in the survey. The household data included detailed information about the respondent family and every individual within the family. The survey questionnaire responses contained detailed information on basic demographics, household, health status, healthcare, employment, household economy

(income, consumption and wealth), health service utilisation and health knowledge. All data were collected by face-to-face computer-aided personal interviews (CAPI) and were entered into Epidata for analysis. Written informed consent was obtained from all participants.

### Exposure and outcome variables
In this study, the main outcome variables included outpatient and inpatient information. Outpatient service utilisation data included the frequency of outpatient visit during the past year (none, one or two or more), place of outpatient visit (ie, private clinics, rural health centres, township hospitals, healthcare facilities in China or others), and average cost for outpatient visits; inpatient service utilisation data included the frequency of inpatient visit during the past 2 weeks (none, one or two or more), reasons for inpatient visit (disease, poisoning and injury, delivery or others), place of inpatient visit (private clinics, rural health centres, township hospitals, healthcare facilities in China or others) and average cost for inpatient visits.

Exposure variables included age, gender, education level (ie, illiterate, 0–5 years of education, ≥6 years and under 14 years old—individuals aged below 14 were categorised as 'under 14 years old' due to ongoing education), ethnicity (ie, Dai, Anni, Wa, Lahu, Kokang, Jingpo, Burman, Blang, etc), occupation (ie, government/military, engineering, business and agriculture, others and under 14 years old—individuals aged below 14 were categorised as 'under 14 years old' due to ongoing education), physical distance from home to healthcare facilities (ie, <5 km, 5–20 km and >20 km), annual household income (ie, US$0 to US$360, US$360 to US$72 and >US$720) and general food security (ie, enough food supply or lacking in food supply).

### Statistical analysis
Selected demographic and socioeconomic variables were considered separately for males and females. The distribution of sociodemographic, unadjusted means with SD for continuous variables and unadjusted proportions for categorical variables were represented. Health service utilisation data including the frequency, reasons and places for inpatient and outpatient visit were separately considered by region and gender. The average costs for inpatient/outpatient visits were distinguished by different types of healthcare facilities.

To analyse the association between physical distance to the nearest healthcare facility and the prevalence of inpatient and outpatient visits, linear regression was performed and adjusted means of frequency to inpatient and outpatient service per thousand population by physical distance was calculated. All linear models were commonly adjusted for covariates, including age, gender, household income and ethnicity.

The association between each exposure variable and inpatient/outpatient utilisations was analysed using logistics regression. OR of using inpatient/outpatient services

were reported within each gender, income level, education level and physical distance to nearest facility. As the population under the age of 14 did not report education level, results were separately considered for population below and above 14 years of age. All statistical analyses were performed using SAS V.9.3, and test results were reported significant at 0.05 level.

## RESULTS

Baseline participants' characteristics stratified by regions are presented in table 1. Of the 4235 participants included in the analyses, 3246 (76.60%) were from the Wa region, and 1638 (50.46%) in Wa region and 500 (50.56%) in SR4 were male, respectively. 41.74% of Wa region's population was under the age of 14, and 32.56% of the Wa population was between 15 and 34 years of age. In comparison, 31.75% and 34.88% of SR4 population were under 14 years and between 15 and 34 years, respectively. 44.31% and 45.91% of Wa and SR4 residents were illiterate, respectively. 29.70% of the Wa population and 73.00% of the SR4 population reported an annual income of above US$720, while a significant portion of the population in Wa region (44.45%) reported an annual household income of less than US$360. In the Wa region, most of the population (82.01%) was of Wa ethnicity, while SR4 was constituted by a more complex variety of ethnicities. Half of the households in SR4 reported a physical distance to the nearest healthcare facilities within 5 km, while about 34.54% of the surveyed households in the Wa region reported a physical distance to the nearest healthcare facilities over 20 km.

Table 2 shows the characteristics of healthcare service utilisation in the study regions. Only 2.16% of Wa residents and 4.65% of SR4 residents reported having received inpatient treatment in the past year. Similarly, only 6.56% of Wa residents and 6.88% of SR4 residents reported having paid outpatient visit in the past 2 weeks. Having disease conditions was the major reason for both inpatient and outpatient service utilisations in the Wa region and SR4. While more residents in the Wa region preferred private clinics for outpatient services (61.03%) and township hospitals for inpatient services (45.71%), more SR4 residents chose healthcare facilities in China for inpatient service (39.13%). Of those who used health services, men were more likely to choose private clinics or health services in China. As shown in figure 1, the average cost for outpatient and inpatient visit in China's health facilities was US$449.09 and US$1414.58, respectively, which was two to three times higher than that in township hospitals (US$205.28 and US$411.37). The average cost for primary healthcare provided by private clinics and rural health centres was much lower, but utilisation of these services remained relatively low.

Figure 2A,B describes the adjusted mean frequency of inpatient/outpatient visits per thousand population

**Table 1** Basic characteristics of 4235 participants by region

| | Wa region (N=3246) | Eastern Shan State SR4 (N=989) |
|---|---|---|
| **Age, years** | | |
| 0–14 | 1355 (41.74) | 314 (31.75) |
| 15–34 | 1057 (32.56) | 345 (34.88) |
| ≥35 | 834 (25.70) | 330 (33.37) |
| **Sex** | | |
| Male | 1638 (50.46) | 500 (50.56) |
| **Education** | | |
| Under 14 years old | 1355 (41.74) | 314 (31.75) |
| Illiterate | 1428 (44.31) | 454 (45.91) |
| 0–5 years | 313 (9.64) | 147 (14.86) |
| ≥6 years | 140 (4.31) | 74 (7.48) |
| **Household income** | | |
| <US$360 | 1443 (44.45) | 82 (8.29) |
| US$360–US$720 | 839 (25.85) | 185 (18.71) |
| >US$720 | 964 (29.70) | 722 (73.00) |
| **Ethnicity** | | |
| Wa | 2662 (82.01) | – |
| Dai | 141 (4.34) | 352 (35.59) |
| Aini | – | 158 (15.98) |
| Others | 443 (13.65) | 479 (48.43) |
| **Occupation** | | |
| Under 14 years old | 1355 (41.74) | 314 (31.75) |
| Government/military | 105 (3.23) | 37 (3.74) |
| Engineering, business and agriculture | 1670 (51.45) | 598 (60.47) |
| Others | 115 (3.57) | 40 (4.04) |
| **Types of drinking water** | | |
| Well water | 17 (0.52) | 140 (14.16) |
| Hose water | 806 (24.83) | 685 (69.26) |
| Others | 2423 (74.65) | 164 (16.58) |
| **Physical distance to nearest healthcare facility** | | |
| <5 km | 1080 (33.27) | 487 (49.24) |
| 5–20 km | 1045 (32.19) | 355 (35.90) |
| >20 km | 1121 (34.54) | 147 (14.86) |
| **Food security** | | |
| Lacking in food supply | 1187 (36.57) | 171 (17.29) |

Values are n (%) unless otherwise specified.
SR4, Special Region 4; WA region, Shan State Special Region 2.

by physical distance. In general, residents living within 5 km paid more outpatient visits to healthcare facilities in the 2 weeks before the survey, compared with residents living 5–20 km and over 20 km (90.06, 54.84 and 54.02 per thousand population per 2 weeks, respectively). In comparison, residents living within

**Table 2** Characteristics of health service utilisation by region and sex

| | Wa region (N=3246) | SR4 (N=989) | P value | Male (N=2138) | Female (N=2097) | P value |
|---|---|---|---|---|---|---|
| Inpatient in the past year | | | <0.001 | | | 0.0180 |
| Yes | 70 (2.16) | 46 (4.65) | | 46 (2.15) | 70 (3.23) | |
| No | 3176 (97.84) | 943 (95.35) | | 1092 (97.85) | 2027 (96.77) | |
| Reason for inpatient among all inpatient population | | | <0.001 | | | 0.0402 |
| Disease | 64 (91.43) | 39 (84.78) | | 45 (97.83) | 58 (82.86) | |
| Poisoning and injury | 2 (2.86) | 1 (2.17) | | 1 (2.17) | 2 (2.86) | |
| Delivery | 4 (5.71) | 6 (13.05) | | – | 10 (14.28) | |
| Place for inpatient among all inpatient population | | | 0.0003 | | | 0.8946 |
| Rural health centres | 1 (1.43) | 3 (6.52) | | 1 (2.17) | 3 (4.29) | |
| Township hospitals | 32 (45.71) | 8 (17.40) | | 14 (30.43) | 26 (37.14) | |
| Private clinics | 19 (27.14) | 6 (13.04) | | 11 (23.91) | 14 (20.00) | |
| Healthcare facilities in China | 13 (18.57) | 18 (39.13) | | 13 (28.26) | 18 (25.71) | |
| Others | 5 (7.15) | 11 (23.91) | | 7 (15.23) | 9 (12.86) | |
| Outpatient in the past 2 weeks | | | 0.7286 | | | 0.2234 |
| Yes | 213 (6.56) | 68 (6.88) | | 132 (6.17) | 149 (7.11) | |
| No | 3033 (93.44) | 921 (93.12) | | 2006 (93.83) | 1948 (92.89) | |
| Place for outpatient among all outpatient population | | | 0.0003 | | | 0.2113 |
| Rural health centres | 15 (7.04) | 6 (8.82) | | 10 (7.57) | 11 (7.38) | |
| Township hospitals | 47 (22.07) | 23 (33.82) | | 30 (22.73) | 40 (26.85) | |
| Private clinics | 130 (61.03) | 23 (33.82) | | 77 (58.33) | 76 (51.01) | |
| Healthcare facilities in China | 9 (4.23) | 14 (20.60) | | 11 (8.33) | 12 (8.05) | |
| Others | 12 (5.63) | 2 (2.94) | | 4 (3.04) | 10 (6.71) | |

Values are n (%) unless otherwise specified. P value refers to comparison between Wa region and SR4, or between males and females.
SR4, Special Region 4; WA region, Shan State Special Region 2.

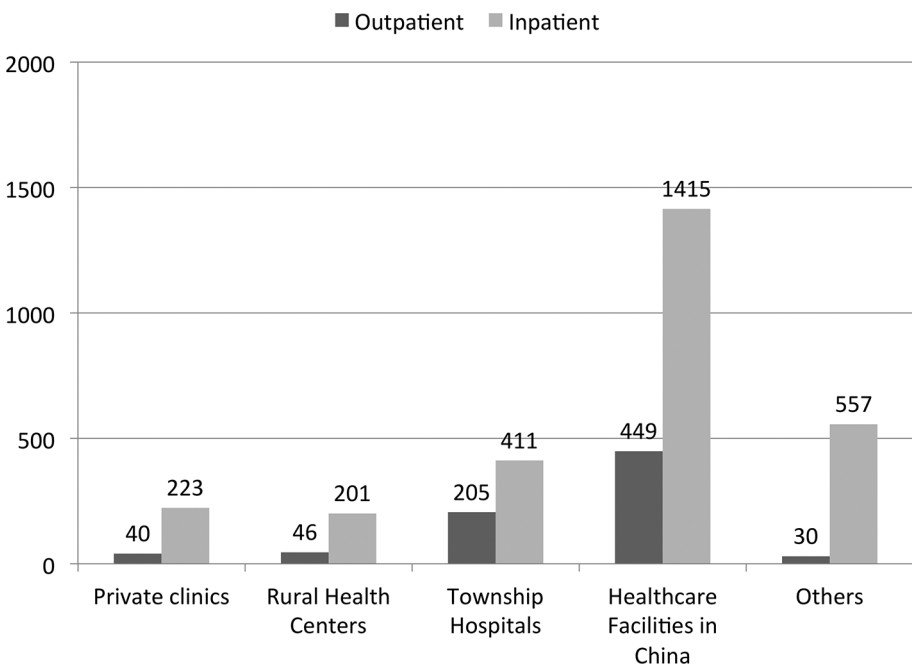

**Figure 1** Average cost for inpatient and outpatient visit in US$.

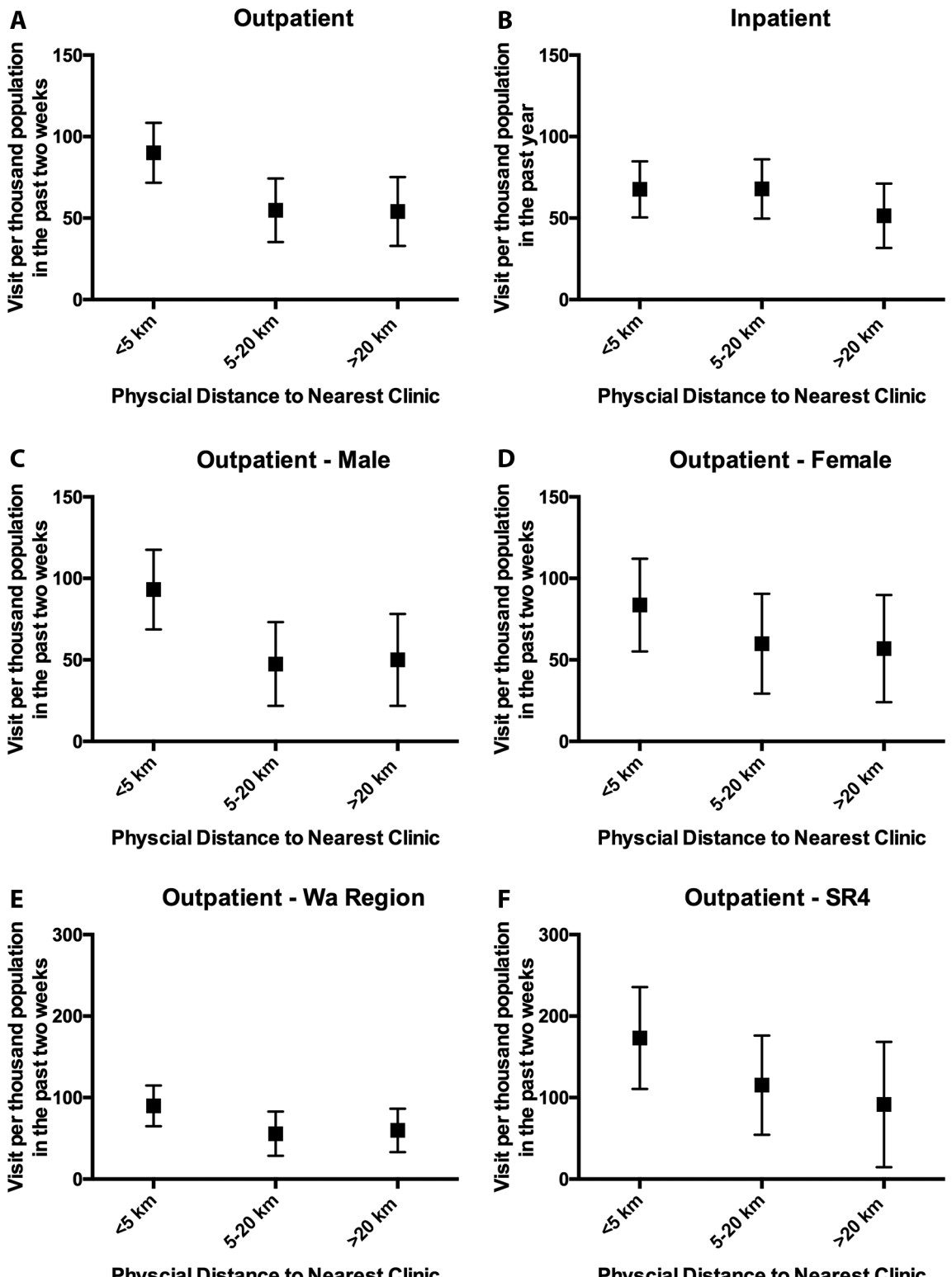

**Figure 2** Adjust means of frequency to inpatient and outpatient service by physical distance*. *Adjusted for age, gender, household income and ethnicity. SR4, Special Region 4; WA region, Shan State Special Region 2.

5 km and between 5 and 20 km used slightly more inpatient services than did residents living over 20 km in the past year. Figure 2C,D illustrates the gender difference in outpatient visits. For men, the average frequency of outpatient service utilisation decreased from 93.15 visits per thousand population for those living within 5 km to the nearest hospital/clinic to 50.01 for those living over 20 km in men; and for women, the average frequency decreased from 83.63 for those living within 5 km to 56.96 in those living over 20 km. As shown in figure 2E,F, SR4 population reported a larger number of outpatient visits within

**Table 3** OR of the association between selected variables and health seeking behaviour in Wa region and Eastern Shan State SR4*

| | Outpatient | | Inpatient | |
|---|---|---|---|---|
| | Age <14 OR (95% CI) | Age ≥14 OR (95% CI) | Age <14 OR (95% CI) | Age ≥14 OR (95% CI) |
| Sex | | | | |
| Male | 1.00 | 1.00 | 1.00 | 1.00 |
| Female | 1.06 (0.66 to 1.72) | 0.86 (0.64 to 1.15) | 1.43 (0.55 to 3.73) | 0.55 (0.35 to 0.84) |
| Household Income | | | | |
| <US$360 | 1.00 | 1.00 | 1.00 | 1.00 |
| US$360–US$720 | 3.43 (1.73 to 6.83) | 1.04 (0.69 to 1.55) | 1.71 (0.51 to 5.76) | 1.53 (0.82 to 2.86) |
| >US$720 | 2.95 (1.48 to 5.86) | 1.43 (0.98 to 2.10) | 1.04 (0.28 to 3.85) | 1.99 (1.09 to 3.64) |
| Distance to nearest healthcare facilities | | | | |
| <5 km | 1.00 | 1.00 | 1.00 | 1.00 |
| 5–20 km | 0.37 (0.21 to 0.66) | 0.67 (0.47 to 0.95) | 0.77 (0.26 to 2.28) | 1.07 (0.66 to 1.73) |
| >20 km | 0.17 (0.07 to 0.38) | 0.79 (0.55 to 1.13) | 0.45 (0.12 to 1.74) | 0.75 (0.43 to 1.32) |
| Education | | | | |
| Illiterate | – | 1.00 | – | 1.00 |
| 0–5 years | – | 0.55 (0.35 to 0.86) | – | 1.12 (0.66 to 1.90) |
| ≥6 years | – | 0.64 (0.36 to 1.15) | – | 0.76 (0.34 to 1.74) |
| Region | | | | |
| Wa | 1.00 | 1.00 | 1.00 | 1.00 |
| SR4 | 0.42 (0.17 to 1.08) | 1.04 (0.64 to 1.68) | 3.35 (0.45 to 25.00) | 1.43 (0.75 to 2.71) |

*All models commonly adjusted for sex, household income, physical distance to nearest healthcare facility, education level and ethnicity.
SR4, Special Region 4; WA region, Shan State Special Region 2.

each distance level than those in Wa region; the mean frequency of outpatient visit for those living within 5 km was 89.85 per thousand population in the past 2 weeks in Wa region, while the corresponding value for SR4 was 173.09. For those living further than 20 km, the mean frequency was 59.77 and 91.57 per thousand population for Wa region and SR4, respectively.

Table 3 shows the association between selected sociodemographic indicators and inpatient/outpatient visits. After adjusting for other covariates, women above 14 were less likely to seek inpatient treatment compared with men of the same age group (OR=0.55, 95% CI 0.35 to 0.84), and slightly less likely to seek outpatient treatment than men (OR=0.86, 95% CI 0.64 to 1.15), while no such association was found among children of both sexes aged below 14 years. Greater distance to the nearest healthcare facilities was associated with a decreased likelihood of inpatient/outpatient utilisations in all age groups. Household income was positively associated with outpatient behaviour. For those aged above 14 years, those with an annual household income of above US$720 were more likely to seek outpatient service (OR=1.43, 95% CI 0.98 to 2.10) compared with those with an income of less than US$360 per year. A similar association was also observed in inpatient visits between the two income groups (OR=1.99, 95% CI 1.09 to 3.64).

As for those aged below 14, a household income between US$360 and US$720 and above US$720 both increased the odds of seeking outpatient services (OR=3.43, 95% CI 1.73 to 6.83, and OR=2.95, 95% CI 1.48 to 5.86, respectively). As for education level, longer years of education was associated with reduced odds of outpatient visits (OR=0.64 and 0.55 for those with an education of longer than 6 years and between 0 and 5 years, respectively).

## DISCUSSION

The present study is the largest and one of the few studies on health inequity in the Myanmar–China border area. Through the analysis of a population-representative survey in the Wa and SR4 regions, we found that health inequity was a major issue leading to a poor health status in the region. Such an inequity in health was attributable to poverty, lack of financially protective measures, geographic barriers and gender-based disparity.

There are several potential limitations of this study: first, the study's cross-sectional nature does not allow for causal inference to provide more robust evidence; second, the interviews were conducted by an international non-governmental organisation (NGO) that provided public health service to the regions, thus potential interviewer bias and respondent's recall bias need to be taken into

account. Nonetheless, the present findings from the population-representative survey provide good insight into service utilisation in Northeastern Myanmar border area, where health data are scarce.

## Inequitable health due to poverty and a lack of financial protection

Our study found that socioeconomic condition, represented by household income, is strongly associated with inpatient and outpatient service utilisation in this region. Besides the fact that generally low income leads to low service utilisation in both inpatient and outpatient services, the financial burden of health services also deters patients from accessing different tiers of healthcare facilities. People with higher household incomes tend to visit healthcare facilities more frequently or use more expensive healthcare facilities in China. In contrast, most of the population with lower household income tend to use private clinics, and in most occasions, small pharmacies, where services are more likely to be provided by uncertified and unqualified health practitioners or pharmacists.

Out-of-pocket expenditure is still a major source of total health expenditure in Northeastern Myanmar border areas, and the proportion is estimated to be higher than that of central Myanmar, which was above 90% in 2014.[9] Considering that most of the households' annual income is less than US$1 per day in Wa region, an outpatient visit to township hospital could cost over half of an annual household income. Besides, the local health system's bias against civilians exacerbates the problem of health inequity. Both Wa region and SR4 cover the outpatient/inpatient fee for military personnel, and reimburse 50% of the fee for soldiers' family members. However, no financial coverage is provided for civilian and children. On the other hand, in the hospitals and rural health centres opened by central Myanmar government, many health professionals charge informal payments or redirect the patients to private hospital/clinics. This poses a financial barrier for civilians in lower socioeconomic status in gaining access to government-funded healthcare services.[8] In both Myanmar government health facilities and ethnic health organisations, high fee-for-service may bring about catastrophic healthcare expenditures, which makes healthcare services unaffordable to patients at the lower end of the socioeconomic spectrum. Tackling the financial barriers is perhaps one of the most crucial and pressing task in improving health service utilisation in Northeastern Myanmar.

## Geographic disparities

Our study found that physical distance to the nearest healthcare facilities could largely influence people's frequency of inpatient and outpatient use in the study regions. The result is consistent with most empirical studies as the patients may have difficulty accessing hospitals and clinics if the physical distance is too great.[10 11] Lack of security due to ongoing armed conflicts in the region, degradation of transportation infrastructure and

unstable external environment, including extreme weather may cause health equity to deteriorate. Nonetheless, our study suggests that geographic distance mainly influences outpatient visits and access to primary healthcare in particular. There is no evidence that physical distance influences inpatient service utilisation in these regions, possibly because inpatient visits are often closely correlated with severe disease conditions in which people must be transferred to health facilities for immediate medical attention.

While the lack of transportation infrastructure—people usually have to walk to the clinics especially in extreme weather conditions—largely obstructs the accessibility to healthcare facilities, uneven geographical distribution of healthcare facilities is also a major factor contributing to the inequity situation. Over 50% of the population does not have access to healthcare facilities within 5 km in both regions. There are only 8 general hospitals, 60 rural health centres and less than 100 private clinics (most of them private pharmacies) in the region serving the 0.6 million population. Most of these healthcare facilities are located in capital regions, leaving the suburban and rural areas in a state of vacuum in terms of access to healthcare services. In the event of an armed conflict, most of the Burmese doctors and private doctors from China would choose to suspend their services and retreat to safer regions. The unbalanced service distribution, exacerbated by armed conflicts, further aggravates the health inequity situation in the ethnic regions of Northeastern Myanmar.

## Gender-based inequities

In the study population of those above 14 years old, women were less likely to seek inpatient treatment compared with men. Women are usually recognised as a particularly vulnerable demographic group as they constitute the majority of internally displaced population and refugees in conflict settings.[8] Rural and ethnic minority women in Myanmar, in particular, could barely achieve equal rights with men. A series of cultural, economic, legal and political factors may perpetuate gender-based inequity to healthcare services in these regions, including but not limited to cultural norms, the low social status of women and their financial dependence on men.[12] Despite international NGO's long-term efforts on maternal and child health, very few women enjoyed hospital-facilitated child delivery, accounting for only 14.28% of all inpatient service for women, which was much lower than Myanmar's national average of 36.2%.[13] It is important to empower women in ethnic regions not only to improve the general health status but also to overcome social and cultural barriers to gender disparity.

This is one of the first studies to explore the different aspects of health inequity, including geographic barrier, gender-based disparity as well as financial burden of health services in the border areas of Northeastern Myanmar. The findings suggest that all these factors strongly restrict local people's access to healthcare services. Tackling health inequity in the Myanmar border

areas requires a better primary healthcare system to fill the gap of geographic disparity, a better financial protection mechanism for the current health system as well as a special focus on women and girls. However, prolonged armed conflicts complicate health service delivery by stressing the region's already weak health infrastructures and worsen the gender inequity problem.

Empirical evidence suggests that countries/regions emerging from war and conflict usually end up having weak or even non-operational health systems, which further expose them to emerging and re-emerging epidemics.[14] The sole focus on vertical programme from international development assistance could serve short-term needs but may in fact further delay the strengthening of the local health system. While the national ceasefire agreement has encountered a huge challenge,[15] the nationwide peace process and the new '21st Century Panglong Conference' suggest new opportunities for health system strengthening in the ethnic regions in Northeastern Myanmar. With additional support and financial resources from national government and international agencies, a comprehensive package of primary healthcare system should be established to reduce health inequity for ethnic peoples.

**Contributors** TK, ZYX and LBL made the statistical analysis and drafted the manuscript; ZSQ and SHL revised the article for important intellectual content and wrote part of the article. All authors read and approved the final manuscript.

**Competing interests** None declared.

**Patient consent** Obtained.

**Ethics approval** Ethical approval was obtained from Zhejiang University.

**Provenance and peer review** Not commissioned; externally peer reviewed.

**Data sharing statement** The data set supporting the conclusion of this article is available from a third party. Please contact the corresponding author to apply to gain access to the relevant data.

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
