## [Reviewer comments · BMJ Open]

ARTICLE DETAILS

TITLE (PROVISIONAL)	Health inequity on access to services in the ethnic minority regions of Northeastern Myanmar: A cross-sectional study
AUTHORS	Tang, Kun; Zhao, Yingxi; Li, Bolun; Zhang, Siqiao; Lee, Sung Hoon

VERSION 1 – REVIEW

REVIEWER	Dr Michael Thiede Scenarium Group GmbH, Germany
REVIEW RETURNED	18-Sep-2017

GENERAL COMMENTS	This is a very good and straightforward study that is presented in a very clear and transparent manner. The discussion is appropriate. Given the setting, the topic is obviously very timely and highly relevant. I am not aware of similar studies that have been performed in the specific geo-cultural context. The discussion is coherent. As the main body of text is written in clear language, I have not ticked "minor revision" with a request for an extra round of language editing. However I see a few very minor glitches in the abstract and suggest that the language in the abstract be polished, as this forms the figurehead of your work.
--

REVIEWER	Davide Rasella Instituto Gonçalo Moniz - IGM. Rua Waldemar Falcão, 121, Candeal - Salvador/BA CEP: 40296-710 Brazil
REVIEW RETURNED	21-Sep-2017

GENERAL COMMENTS	This is a very interesting and relevant paper which evaluates with an appropriate statistical analysis the inequalities in the access to healthcare in Northeastern Myanmar. A journal such as BMJ Open can be the right place for this paper, having this manuscript relevance in particular for policymakers and epidemiologist. I suggest only some minor revisions: 1. The title suggests a study on health inequities instead of a study on inequities on the access to healthcare.2. You mention infant mortality in Eastern Myanmar but not any other data about the country, it would be useful to mention few health statistics of the country (even from http://www.who.int/gho/countries/mmr.pdf?ua=1) or of the region if available.
---

	3. Line 133. I assume 360 USD (and 720) is per year, but I do not see it specified. Moreover considered that almost 50% of the population in the We region is below this threshold (while in the SR\$ is 8%), I suggest, if possible, to conduct a sensitivity analysis, or a complementary analysis, with lower thresholds. The relation income – access to healthcare (outpatient and inpatient) should be non-linear, so if you use a threshold of 150 or 200 for example you should find a – possibly by far – higher OR. This is relevant because still probably represents 10-30% of the population (I do not know the distribution). 4. I would suggest substituting the term “nutritional status” with something as food security; it seems more adequate considering the measure you use to evaluate it. 5. As far as I can see Wa region and SR4 region have very different socioeconomic profiles, and you correctly have stratified the analyses per region. But when you study the exposure factors you use all the sample, so somehow you mix two population which could have different population-specific effects worthy to be shown. In particular the Wa region, could show some more extreme exposure effects that what is shown when you analyze them together. If you have done this in your data analysis, could be interesting to mention it as sensitivity or complementary analysis.
--	--

VERSION 1 – AUTHOR RESPONSE

Reviewer: 1

Reviewer Name: Dr Michael Thiede

Institution and Country: Scenarium Group GmbH, Germany

Please state any competing interests: None declared

Comment: This is a very good and straightforward study that is presented in a very clear and transparent manner. The discussion is appropriate. Given the setting, the topic is obviously very timely and highly relevant. I am not aware of similar studies that have been performed in the specific geo-cultural context. The discussion is coherent.

As the main body of text is written in clear language, I have not ticked "minor revision" with a request for an extra round of language editing. However I see a few very minor glitches in the abstract and suggest that the language in the abstract be polished, as this forms the figurehead of your work.

Response: Thank you for your comments and encouragement, we have polished our language by a native English speaker as you kindly requested.

Reviewer: 2

Reviewer Name: Davide Rasella

Institution and Country: Instituto Gonçalo Moniz - IGM. Rua Waldemar Falcão, 121, Candeal - Salvador/BA CEP: 40296-710, Brazil

Please state any competing interests: None Declared

1. The title suggests a study on health inequities instead of a study on inequities on the access to healthcare.

Response: We have revised our title to fit editorial requirement and your suggestion

2. You mention infant mortality in Eastern Myanmar but not any other data about the country, it would be useful to mention few health statistics of the country (even from <http://www.who.int/gho/countries/mmr.pdf?ua=1>) or of the region if available.

Response: We have added the infant mortality and under 5 mortality of Myanmar in 2013 – Please refer to page 3 line 26 and Reference #7

3. Line 133. I assume 360 USD (and 720) is per year, but I do not see it specified.

Response: Thank you for the kind reminder, we have added in page 4 line 38.

- Moreover, considered that almost 50% of the population in the We region is below this threshold (while in the SR\$ is 8%), I suggest, if possible, to conduct a sensitivity analysis, or a complementary analysis, with lower thresholds. The relation income – access to healthcare (outpatient and inpatient) should be non-linear, so if you use a threshold of 150 or 200 for example you should find a – possibly by far – higher OR. This is relevant because still probably represents 10-30% of the population (I do not know the distribution).

Response: We have done an additional analysis specifically for Wa region residents aged above 14, and set a separate group for household income below 150 USD, the result was similar as shown below. Also considering that there are few people with annual income below 150 USD in SR4, we have therefore chosen not to present the result in the manuscript to keep categories consistent across the two regions.

Income group	Outpatient	Inpatient
<150 USD	1.00	1.00
150-360 USD	0.795 (0.441-1.433)	0.790 (0.270-2.311)
360-720 USD	1.021 (0.639-1.633)	1.678 (0.777-3.627)
>720 USD	1.427 (0.923-2.205)	2.299 (1.096-4.819)

4. I would suggest substituting the term “nutritional status” with something as food security; it seems more adequate considering the measure you use to evaluate it.

Response: Revised as requested, please refer to page 4 line 39 and table 1

5. As far as I can see Wa region and SR4 region have very different socioeconomic profiles, and you correctly have stratified the analyses per region. But when you study the exposure factors you use all the sample, so somehow you mix two population which could have different population-specific effects worthy to be shown. In particular the Wa region, could show some more extreme exposure effects that what is shown when you analyze them together. If you have done this in your data analysis, could be interesting to mention it as sensitivity or complementary analysis.

Response: Thank you for pointing out the issue. To provide a better estimate of the odds ratios, we have added “region” as a confounding variable into the regression model, and presented the region-adjusted results in Table 3. This will minimize the influence of region to the association. Besides, the geopolitical condition of Wa region and SR4 region are similar, both were governed by ethnic minority authorities and health services were provided in parallel by Myanmar’s Ministry of Health and Sports as well as local ethnic health authorities, thus we believe practically, it is reasonable to pool the data of two regions together while adjusting for the “cluster” in the regression models.